# Does Occupational Exposure to Chemicals/Carcinogens Affect the Hematological Parameters of Workers?

**DOI:** 10.3390/jcm13216317

**Published:** 2024-10-22

**Authors:** Giuseppe La Torre, Maria Vittoria Manai, Luca Moretti, Francesca Vezza, Massimo Breccia, Arturo Cafolla, Gabriele Bernardinetti, Gaetano Leo, Katia Stefanantoni, Antonio Pavan, Sabina Sernia

**Affiliations:** 1Department of Public Health and Infectious Diseases, Sapienza University of Rome, 00161 Rome, Italy; mariavittoria.manai@uniroma1.it (M.V.M.); luca.moretti@uniroma1.it (L.M.); francesca.vezza@uniroma1.it (F.V.); arturo.cafolla@uniroma1.it (A.C.); bernardinetti.1898329@studenti.uniroma1.it (G.B.); leo.1526659@studenti.uniroma1.it (G.L.); sabina.sernia@uniroma1.it (S.S.); 2Department of Translational and Precision Medicine, Sapienza University of Rome, 00161 Rome, Italy; massimo.breccia@uniroma1.it; 3Department of Clinical and Molecular Medicine, Sapienza University of Rome, 00161 Rome, Italy; katia.stefanantoni@uniroma1.it (K.S.); antonio.pavan@uniroma1.it (A.P.)

**Keywords:** exposure, chemicals, carcinogens, lymphocytes, red cells, monocytes

## Abstract

**Background/Objectives:** Chemical and carcinogenic risk in workplace are linked to an increase of the incidence of cancer in exposed workers. The aim of this observational study was to identify an association between exposure to chemical risk and changes in hematochemical parameters of workers exposed. **Methods:** The study examined blood count parameters performed on 4523 employees of Sapienza University of Rome from September 2019 to February 2020. A total of 1402 workers in research laboratories, exposed to chemical risk and carcinogens, was compared with a cohort of 3121 blood donors, in apparently good health, matched for age, sex, and period of blood collection. **Results:** Multivariate analysis demonstrated acrylamide is the most frequently associated chemical with significant changes between the groups. It is related to an increase in monocytes and MPV, a reduction of lymphocytes, platelets, and red series values, while MCH remained unchanged. Formaldehyde reveals the same changes as acrylamide, with the exception of the percentage of lymphocytes, red blood cells, and HB, which remain unchanged. Other chemicals and carcinogens show variations for few blood count parameters. **Conclusions:** The present study found out a significant change in blood count values in workers exposed to acrylamide and formaldehyde. It also showed an increase in the mean monocyte value in employees exposed to carcinogenic risk, compared to non-exposed, with the exception of Trypan Blue. The implementation of efficient health surveillance remains of critical importance. These data should be compared with those from follow-up studies.

## 1. Introduction

Chemicals are a fundamental and enduring part of our society. Most of our wealth comes from the exploitation of chemical reactions to produce everything, from pharmaceuticals to water purifiers, from paints to plastics [1]. Sales of chemicals in 2007 amounted to 537 billion euros within the EU, and chemicals, plastics and rubber industry creates around 3.2 million jobs in more than 60,000 companies. However, exposure to chemicals in the workplace can pose significant risks to human health, including carcinogenesis, the process by which normal cells are transformed into cancerous cells [1]. Hence the importance of understanding and reducing these risks to safeguard the health and well-being of workers [1].

Chemicals can lead to undesirable health, environmental, or safety effects, and this has led legislation to regulate their use [1] Title IX, Chapter I, of Legislative Decree 81/2008 deals with the assessment of risk from exposure to hazardous chemical agents in the workplace [2]. The assessment should consider the main routes of introduction of chemical agents into the human body, in particular the respiratory route, by inhalation, and the dermal route, by absorption.

In the case of work activities involving exposure to several hazardous chemical agents, the risk resulting from the combination of all the chemical agents should be assessed. If a new activity, involving hazardous chemical agents, is started, a risk assessment must be carried out beforehand, and the relevant prevention measures implemented.

The “MoVaRisCh” model [3], approved by the technical groups of Emilia-Romagna, Tuscany, and Lombardy regions, in application of Title IX Chapter I of Legislative Decree 81/08, is used for the assessment of chemical risk within working companies, which allows the assessment of chemical risk for the health of workers in accordance with Article 223 of Legislative Decree 81/08 [2]. Basically, the model is used, first of all, to identify the health risk and, if necessary, through environmental and personal sampling, for the classification above or below the threshold of irrelevant health risk.

Workers are exposed to a range of chemical agents whose hazardousness is classified according to their properties: chemical–physical (explosive, flammable, etc.), eco-toxicological (hazardous to the environment), and toxicological (corrosive, irritant, harmful) [4]. Depending on the specific effects on human health, toxic substances are further divided into carcinogenic substances (substances or preparations that can cause cancer in humans or increase frequency, as benzene), mutagenic substances, that can cause hereditary genetic defects in humans or increase frequency, such as acrylamide.

The use of a chemical substance does not, in itself, necessarily constitute an actual risk to health, as it depends only on the toxicological characteristics of the substance [4] and on the manner of contact that takes place during the course of the work activity [4]. For this reason, chemical risk is linked to the exposure to a given quantity of a substance to which a subject is actually exposed and the relative period of exposure, taking into account the possible routes of penetration into the body.

Absorption of a chemical agent corresponds with its passage from the external environment to the systemic circulation within an organism [4]. In workplace, absorption mainly occurs through three routes: air, through which gases, vapors, aerosols, fumes, and dust can locally damage the respiratory tract; skin, by which liquids and gases pass the stratum corneum, by passive diffusion, and reach the dermis (excoriations, inflammation, and thinning of the skin facilitate the absorption of chemicals); and gastroenteric route, which is related to the swallowing of aerosol-contaminated saliva [4].

Exposure to chemical compounds occurs in various working environments, including research and health laboratories, which are characterized by the extreme variability of the compounds used and the frequent turnover of young researchers.

The main chemical agents to which workers at Sapienza University of Rome are exposed are: benzene, acrylamide, trichloroethylene, potassium dichromate, formalin, formaldehyde, colchicine, cisplatin, chloramphenicol, ligroin, hexavalent chromium, trichloroethylene, nickel sulphate, diamino benzidine, propylene oxide, trypan blue, cadmium chloride hydrate, cobalt, doxorubicin, and epirubicin.

In literature, chemical hazards have been explored in detail in a variety of industrial and agricultural occupational settings [5,6,7,8,9], while in research or healthcare laboratories, attention has been particularly focused on biological hazards [5,10,11,12,13].

The aim of this study is to identify whether there is an association between exposure to chemicals and the occurrence of blood count alterations in employees of Sapienza University of Rome exposed to chemical risk, comparing the results with those of a group of non-exposed people.

## 2. Materials and Methods

### 2.1. Study Design

This cross-sectional study was conducted between 15 September 2019 and 15 February 2020 on a group of workers exposed to chemicals or carcinogens in ‘Sapienza’ University of Rome laboratories and a group of not exposed (blood donors at the Transfusion Centre of the Teaching Hospital Umberto I of Rome).

### 2.2. Data Collection

The study took into consideration the parameters of the hemochromocytometric examination performed on subjects, in apparent good health, working at ‘Sapienza’ University of Rome, exposed to airborne chemical and carcinogenic hazards and undergoing periodic visits at the Centre for Occupational Medicine (CMO) of the same university.

A total of 973 of the 1039 subjects/workers of research laboratories, potentially exposed to chemical risk, observed in the period 15 September 2019–15 February 2020, were taken into account. The 90% (876) of these were most frequently exposed to ‘chemical risk’ because they belonged to the Departments of Chemistry, Chemistry and Technology of Pharmaceuticals, Biochemistry, Pharmacology, and Biotechnology (Table 1).

The inclusion criteria were:age between 21 and 69 years;apparent good health.

Subjects, fasting for at least eight hours, underwent peripheral venous blood sampling, collected in 2.7 mL Vacutainer in EDTA, between 8.30 and 11.30 a.m. The samples for the determination of hemochromocytometric parameters, kept under refrigerated conditions, were processed within five hours of collection.

The hemochromocytometric parameters of the subjects enrolled were compared with the parameters recorded in a cohort of 3600 blood donors, afferent from the 15th September 2019 to the 15th February 2020 to the Transfusion Centre of the Policlinico Umberto I in Rome.

Blood donors were frequency matched with exposed subjects/workers by age, sex, and collection period. The following parameters were taken into account: white blood cells (×10^3^/mmc), absolute value of lymphocytes (×10^3^/mmc), polymorphonucleates (×10^3^/mmc), monocytes (×10^3^/mmc), red blood cells (×10^3^/mmc), hemoglobin gr/dL, hematocrit%, mean corpuscular volume femtolitre, mean corpuscular hemoglobin content (MCH) measured in picograms, mean corpuscular hemoglobin concentration% (MCHC), number of platelets per blood volume (PLTS) × 10^3^/mmc, mean platelet volume (MPV) femtolitres.

Particularly, the reference values used by the laboratory were respectively: leukocytes 4.0–11.0 × 10^3^/mm^3^ of blood; lymphocytes 25–50%; monocytes 2–11%; granulocytes neutrophils 50–80%; granulocytes eosinophils 0–5%; granulocytes basophils 0–2%. No data has been collected on the immunological aspects that could increase knowledge.

All the subject involved agreed on the treatment of personal data, even sensible ones, used in anonymous and collective way, with scientific modality and aims, according to the principles of the Helsinki Declaration.

### 2.3. Data Analysis

Data were analyzed by using SPSS, v27.

The statistical analysis, performed to evaluate chemical risk, carried out frequency tables, statistical significance tests and regression analysis. The differences between the two groups, for quantitative variables, were assessed by means of analysis of variance (ANOVA). Then, every quantitative variable was used as a dependent variable in a multiple linear regression model, in which the independent variables were:–exposure to chemical/carcinogenic risks;–female sex;–age.

The results of the regression analysis are presented as standardized β coefficient (*p*-value). A positive value of the coefficient indicates a direct association, while a negative one an inverse association. The quality of fit of the model was assessed using the coefficient of determination R^2^. Finally, multivariate analysis was conducted by stratifying the sample by age (<50 or >50 years old) and by gender (male and female). The analysis was conducted using SPSS software for WINDOWS, 27.0.

The statistical analysis, performed to evaluate carcinogenic risk, also carried out frequency tables, statistical significance tests, and regression analysis. The conformity of the variables to the normal distribution was examined using visual box-plot. Continuous variables were presented as mean ± standard deviation (SD). The ANOVA test was used to perform analysis of variance to analyze the differences between the means of the blood count parameters between the groups. A post-hoc Bonferroni test was then performed to demonstrate which specific groups were significantly different from each other. The quality of fit of the model was assessed using the coefficient of determination R^2^. Finally, multivariate analysis was conducted by stratifying the sample by age.

A *p*-value of <0.05 was accepted as the statistical significance level.

## 3. Results

### 3.1. Characteristics of the Participants

The total number of participants was 4523: 356 were exposed to carcinogens (exposed group), 1046 were exposed to chemicals (second exposed group), and 3121 were in the control group. The mean age was 41.67 ± 1459 years for the exposed to carcinogens, 39.48 ± 14.11 years for the exposed to chemicals, and 40.68 ± 13.87 for non-exposed. Women constituted 61.5% of the exposed to carcinogens (n = 219), 61.8% of the exposed to chemicals (n = 646) and 60.6% of the control group (n = 1890). Age was significant only in exposed to chemicals and non-exposed (*p* = 0.017); there was no significant differences between the groups in terms of gender and age between exposed to carcinogens and non-exposed (Table 1). All the donors in the control group are healthy. Our donor center does not collect data on work history, but we do have a registry on carcinogens in Italy, the SIERP, which is an advanced tool for recording and analyzing the data flow provided for by Article 243 of Legislative Decree No. 81/2008 on occupational exposure registers to carcinogens in Italy. This is compiled on the INAIL platform and allows us to have information on how many people are exposed to carcinogenic risk according to the region they belong to. Knowing that the population in Lazio is exposed to this range, we can conclude that even if some donors might be exposed, this would not affect the *p*-value of <0.001.

Since no differences were found in hemochromocytometric parameters between exposed to chemicals and exposed to carcinogens, the following results show the comparisons between Exposed to chemicals and not exposed (blood donors), and between exposed to carcinogens and not exposed.

### 3.2. Comparison of Average Blood Count Parameters Between Chemically Exposed and Not-Exposed Subjects

The mean values of the hemochromocytometric parameters of workers exposed to chemical risks were compared with those of the non-exposed. The significance of the differences found was assessed using the ANOVA test (Table 2).

This analysis showed a significant increase in the group of exposed workers in the average values of white blood cells (6.8 × 10^3^/µL + −1.8 vs. 6.4 × 10^3^/µL + −1.5—*p* < 0.001), monocytes both in absolute number (0.5 ×10^3^/µL + −0.2 vs. 0.4 × 10^3^/µL + −0.1—*p* < 0.001) and in percentage (7.9% + −1.9 vs. 6.7% + −1.9—*p* < 0.001), platelets (238. 5 × 10^3^/µL + −64.4 vs. 229.5 × 10^3^/µL + −51.7—*p* < 0.001) and their volume (9.8 fL + −30.7 vs. 8.2 fL + −8.6—*p* = 0.008); a significant reduction in the values of lymphocyte percentage (33.8% + −7.5 vs. 36.8% + −7.5—*p* < 0.001), polymorphonuclear percentage (54. 7% + −8.7 vs. 56.4% + −8.1—*p* < 0.001), hemoglobin (13.9 d/dL + −2.4 vs. 14.2 g/dL + −1.2—*p* < 0.001), hematocrit (41.7% + −5.8 vs. 43.0% + −3.8—*p* < 0.001), mean corpuscular volume (87.8 fL + −8.1 vs. 90.3 fL + −4.2—*p* < 0.001), mean corpuscular hemoglobin content (29.4 pg + −2.2 vs. + −29.8 pg + −1.6—*p* < 0.001), mean cellular hemoglobin concentration (33.1 g/dL + −2.1 vs. 32.9 g/dL + −1.0—*p* = 0.002); however, there were no significant differences in the absolute number of lymphocytes, polymorphonucleates, and red blood cells.

### 3.3. Comparison of Average Haemochromocytometric Parameters Between Carcinogen-Exposed and Not-Exposed Subjects

The mean values of the hemochromocytometric parameters of the carcinogen-exposed workers were compared with those of the non-exposed and the significance of the differences found was assessed with the ANOVA test (Table 3).

In the exposed to carcinogens group, significantly higher mean values of monocytes were found both in absolute number (0.5 × 10^3^/µL + −0.2 vs. 0. 4 × 10^3^/µL + −0.2—*p* < 0.001) and in percentage (8% + −1.7 vs. 6.7 + −2.6—*p* < 0.001), in the number of white blood cells (6.8 × 10^3^/µL + −1.7 vs. 6.4 × 10^3^/µL + −1.9—*p* < 0.001), and red blood cells (4.87 × 10^6^/µL + − 1.18 vs. 4.76 × 10^6^/µL + − 0.43—*p*= 0.002), mean corpuscular hemoglobin content (30.9 pg + −20.1 vs. 29.8 pg + −1.6—*p* = 0.003); mean cellular hemoglobin concentration (33.6 g/dL + −16.9 vs. 32.9 g/dL + −1.0—*p* = 0.024) and mean platelet volume (10.4 fL + −37.3 vs. 8.2 fL + −8.6—*p* = 0.005). The same comparison showed significantly lower values for the percentage of polymorphonucleates (54.1% + −8.6 vs. 56.7% + −12.2—*p* < 0.001) and lymphocytes (34.1% + −7.4 vs. 37.0% + −10.7—*p* < 0.001) and the mean corpuscular volume of red blood cells (86.6 fL + −10.9 vs. 90.3 fL + −4.2—*p* < 0.001).

On the other hand, no statistically significant differences were found between exposed and unexposed in the mean absolute number of polymorphonucleates, lymphocytes, and platelets, hemoglobin concentration, and hematocrit.

### 3.4. Comparison of Average Haemochromocytometric Parameters Between Each Individual Carcinogen-Exposed and Non-Exposed Subjects

Exposure to each individual carcinogen was then assessed in comparison to the non-exposed, by comparing the average blood counts and evaluating the significance of the differences using the ANOVA test (Table 4).

Those exposed to acrylamide (125 subjects) had a significantly higher level of monocytes, in both absolute number (0.53 × 10^3^/µL + −0.16 vs. 0.37 × 10^3^/µL + −0.14—*p* = 0.000) and percentage (7.96% + −1.65 vs. 6.66% + −1.88—*p* = 0.000), mean cellular hemoglobin concentration (33.29 g/dL + −1.21 vs. 32.95 g/dL + −1—*p* = 0.013), platelets (248.25 × 10^3^/µL + −64.85 vs. 230.79 × 10^3^/µL + −52.3—*p* = 0.021) and their mean volume (8.47 fL + −1.1 vs. 7.85 + −0.77—*p* = 0.000); the mean values of lymphocyte percentage (33.7% + −7.3 vs. 36.8% + −7.5—*p* = 0.000), red blood cells (4.61 × 10^6^/µL + −0.4 vs. 4.77 × 10^6^/µL + −0.43—*p* = 0.001), hemoglobin (13.42g/dL + −1.14 vs. 14.18g/dL + -1.17—*p* = 0.000), hematocrit (40.34% + −3.33 vs. 43.05% + −3.77—*p* = 0.000) and mean corpuscular volume (87.76 fL + −6.22 vs. 90.34 fL + −4.26—*p* = 0.000) were significantly lower than in non-exposed persons; there were no statistically significant differences in absolute and percentage numbers of polymorphonucleates, absolute numbers of lymphocytes and white blood cells, mean corpuscular hemoglobin content.

Those exposed to benzene (34 subjects) had a significantly higher level of the number of white blood cells (7.37 × 10^3^/µL + −2.13 vs. 6.4 × 10^3^/µL + −1.53—*p* = 0.011) and monocytes (0.55 × 10^3^/µL + −0.21 vs. 0.37 × 10^3^/µL + −0.14—*p* = 0.000); no significant differences were found in polymorphonucleates, both in absolute numbers and percentages, lymphocytes, both in absolute numbers and percentages, the percentage of monocytes, red series parameters (GR, Hb, Ht, MCV, MCH, MCHC), and platelets (PLTS and MPV).

Those exposed to formaldehyde (47 subjects) had significantly higher levels of monocytes, both in absolute number (0.51 × 10^3^/µL + −0.15 vs. 0.37 × 10^3^/µL + −0.14—*p* = 0.000) and percentage (8.02% + −1.67 vs. 6.67 + −1.88—*p* = 0.000), mean cellular hemoglobin concentration (33.49 g/dL + −1.18 vs. 32.95g/dL + −1—*p* = 0.015), mean platelet volume (8.72 fL + −1.01 vs. 7.85 fL + −0.77—*p* = 0.000) and significantly lower values of the hematocrit (40.82% + −3.8 vs. 43.05% + −3.78—*p* = 0.002); no significant differences were revealed in the mean total white blood cells, polymorphonucleates, both in absolute number and percentage, lymphocytes, both in absolute number and percentage, red blood cells, hemoglobin, mean corpuscular hemoglobin content, and platelets.

Those exposed to potassium dichromate (16 subjects) recorded significantly higher values of monocytes, both in absolute number (0.58 × 10^3^/µL + −0.11 vs. 0.37 × 10^3^/µL + −0.14—*p* = 0.000) and percentage (8.2% + −1.48 vs. 6.67 + −1.88—*p* = 0.037); no significant differences were found in the mean values of total white blood cells, polymorphonucleates, both in absolute numbers and percentages, lymphocytes, both in absolute numbers and percentages, red series parameters (GR, Hb, Ht, MCV, MCH, MCHC), and platelets (PLTS and MPV).

Those exposed to ligroin (14 subjects) recorded significantly higher values of the absolute number of monocytes (0.53 × 10^3^/µL + −0.17 vs. 0.37 × 10^3^/µL + −0.14—*p* = 0.001) and platelets (284.79 × 10^3^/µL + −154.53 vs. 230.79 × 10^3^/µL + −53.3—*p* = 0.010); no significant differences were found in the mean values of total white blood cells, polymorphonucleates, both in absolute numbers and percentages, lymphocytes, both in absolute numbers and percentages, the percentage of monocytes, the red series parameters (GR, Hb, Ht, MCV, MCH, MCHC), and platelets.

Those exposed to trypan blue (9 subjects) had significantly higher mean platelet volume (8.73 fL + −1.3 vs. 7.85 fL + −0.77—*p* = 0.033) and significantly lower hemoglobin (12.82 g/dL + −1.1 vs. 14.18 g/dL + −1.17—*p* = 0.021) and hematocrit (38.96% + −3.26 vs. 43.05% + −3.77—*p* = 0.040); no significant differences were found in the white series values (total GB, PMN both absolute number and percentage, lymphocytes both absolute number and percentage, monocytes both absolute number and percentage), the number of red blood cells and their mean corpuscular volume, mean cellular hemoglobin concentration, mean corpuscular hemoglobin content, and the number of platelets.

Those exposed to trichloroethylene (8 subjects) had significantly higher values of monocytes, both in absolute number (0.57 × 10^3^/µL + −0.17 vs. 0.37 × 10^3^/µL + −0.14—*p* = 0.002) and percentage (9.12% + −3.36 vs. 6.67 + −1.88—*p* = 0.000); significantly lower than the percentage of polymorphonucleates (45.% + −8.8 vs. 56.52% + −8.1—*p* = 0.008); no significant differences were found with regard to the values of the total number of white blood cells and polymorphonucleates, of lymphocytes, both in absolute numbers and percentages, of the parameters related to the red series (GR, Hb, Ht, MCV, MCH, MCHC) and to platelets (PLTS and MPV).

In the category “exposed to other carcinogens”, exposures to infrequent carcinogens and exposures to unspecified carcinogens were included and lumped together.

In the latter class, significantly higher values of monocytes were found, both in absolute number (0.54 × 10^3^/µL + −0.18 vs. 0.37 × 10^3^/µL + −0.14—*p* = 0.002) and percentage (8.1% + −3.36 vs. 6.67 + −1.88—*p* = 0.000); significantly lower than the percentage of polymorphonucleates (53.78% + −8.03 vs. 56.52% + −8.1—*p* = 0.028), the mean corpuscular volume of red blood cells (88.32 fL + −6.4 vs. 90.34 fL + −4.24—*p* = 0.000), the mean cellular hemoglobin concentration (29.2 pg + −2.24 vs. 29.77 pg + −1.6—*p* = 0.023) and the mean platelet volume (8.44 fL + −1.04 vs. 7.85 fL + −0.77—*p* = 0.000); no significant differences were found with regard to the values of the total number of white blood cells and polymorphonucleates, lymphocytes, both in absolute number and percentage, the number of red blood cells, the value of hemoglobin, hematocrit, mean corpuscular hemoglobin content, and the number of platelets.

## 4. Discussion

Chemical agents [14,15,16], along with physical agents [17,18] and psychosocial factors [19,20], can be disruptive and interfere with organic homeostasis, producing health effects [14]. The purpose of the present study was to identify whether there is an association in individuals exposed to the chemical hazard and the occurrence of alterations in blood counts, compared with the unexposed group. The results showed that age was significant only in exposed and non-exposed (*p* = 0.017) to chemicals; there was no significant differences between the groups in terms of gender and age between exposed and non-exposed to carcinogens. Furthermore, acrylamide was the chemical compound most frequently associated with significant changes between the two groups, followed by formaldehyde. Acrylamide, in particular, increases monocytes in both absolute value and percentage (*p*-value < 0.001) and MPV (*p*-value < 0.001), while decreasing the percentage of lymphocytes (*p*-value < 0.001), platelet number (*p*-value = 0.002), and red series values (GR *p*-value = 0.002; HB *p*-value < 0.001: HT *p*-value < 0.001; MCV *p*-value < 0.001;); MCH remains unchanged. Formaldehyde shows the same changes as acrylamide, except for the percentage of lymphocytes, red blood cells, and HB, which remain unchanged. The other carcinogens examined show changes for a few CBC parameters. The only significant finding shared by all carcinogens, except for Trypan Blue, is the absolute increase in the number of monocytes (*p*-value < 0.001 for acrylamide, benzene, formaldehyde, K-chromate, and other carcinogens; *p*-value = 0.002 for ligroin and trichloroethylene). In the literature, only changes in CBC are found in animal experiments with acrylamide, unlike our study, in which changes in CBC parameters are shown for all blood cell populations. Acrylamide was in fact classified as a rodent carcinogen and probable human carcinogen by the International Agency for Research on Cancer in 1994 [21] because of its carcinogenicity in rodents and the similarity between the way it is metabolized in rodents and humans. The mechanism by which acrylamide causes cancer in laboratory animals and in humans is still unclear. Currently, the genotoxic action of glycidamide, an epoxide metabolite of acrylamide, is considered the mechanism of carcinogenic action in acrylamide risk assessments [22]. Extensive in vitro and in vivo animal studies have shown that acrylamide, mainly after metabolic conversion to glycidamide by the enzyme cytochrome P4502E1 (CYP2E1), causes chromosomal damage (aberrations, micronuclei, aneuploidies) and mutagenic effects [23]. For example, in the study by Raju et al., anemia and thrombocytopenia as well as increased leukocyte counts were observed following acrylamide administration in mice [24]. In addition, the results described by Grzybowska [25] indicate a decrease in porcine granulocytic cell line activity under the influence of AA. Only high doses of acrylamide caused an increase in the number of neutrophil granulocytes and banded basophils and the appearance of hypersegmented granulocytes in pigs. Low doses of AA led to statistically significant increases in eosinophil numbers; however, despite these increases, total granulocyte counts decreased significantly in both experimental groups to resemble bone marrow hypoplasia in the granulocytic cell line. The results undoubtedly indicate that regardless of animal species (pig, rat, or mouse), AA influences hematopoiesis processes and may have a stimulatory or inhibitory effect on certain cell lines depending on the species studied. A study by Hammad et al. [26] on rats showed an increase in the number of white blood cells in all acrylamide-treated groups. The authors of the present publication observed a decrease in the total number of granulocytes in pigs. In humans, on the other hand, it was seen, particularly in Collins’ study [27], that despite acrylamide exposure, there were similar levels of mortality as in other working populations. No significant excess mortality was observed in 26 cancer sites examined in the exposed group. Seventy-two cancer deaths occurred among the 2293 people exposed to acrylamide, compared with 73.4 expected cancer deaths. In contrast, this study points out that, in an animal study, the central nervous system and the reproductive organ were the most affected sites. This is also emphasized in Granath’s study [28], namely, that with the exception of a weak significance for an increased incidence of pancreatic cancer, there is “little evidence for a causal relationship between acrylamide exposure and mortality from any cancer site”.

Another compound widely used among the workers in this study is formaldehyde. Formaldehyde is an organic compound that, at room temperature and standard atmospheric pressure, is in the form of a colorless, pungent, and irritating gas that is extremely volatile and highly soluble in water. It is present as a natural product in many living systems, in the environment, in some foods, and in the bodies of mammals, including humans, as a product of oxidative metabolism. Formaldehyde is a known occupational carcinogen: it is a sensory irritant compound, recognized especially for sensitive individuals [29], present in many different work scenarios. In fact, formaldehyde is widely used in many production processes and health applications due to its chemical and physical characteristics and broad-spectrum microbicidal activity [30]. The International Agency for Research on Cancer (IARC) [31] has identified three main occupational scenarios, in which workers may be exposed to formaldehyde at airborne concentrations significantly above indoor and outdoor background levels: (1) the production of formaldehyde and/or its solutions; (2) the manufacture of formaldehyde-containing products or during their use; and (3) the combustion of formaldehyde-generating products. Thus, operators of industrial production processes (resins, plastics, semi-finished wood products, furnishings, and textiles), professionals in pathological anatomy laboratories, veterinarians, taxidermists, breeders, carpenters, industrial laundries, firefighters, beauticians, and printers are the categories at greatest risk of formaldehyde exposure [32,33].

In 2012, it was stated that there was strong but insufficient evidence of a causal relationship with leukemia and formaldehyde exposure [34], and with irritant or neoplastic diseases [35]. Two studies emphasized the relationship between formaldehyde exposure and lymphohematopoietic cancers, but no association was observed for all leukemias [36], except for a small and weak association with non-Hodgkin’s Lymphoma [37] and myeloid leukemia [38]. This is consistent with the results of other previous studies [39,40]. For formaldehyde exposure, on the other hand, it is shown that occupational exposure to formaldehyde can cause alterations in white blood cell counts in exposed male health care workers compared to unexposed male health care workers [14]. Specifically, in male subjects exposed to formaldehyde, the mean lymphocyte values were significantly higher than in unexposed subjects (33.08% vs. 30.1%; *p* = 0.04); in exposed male subjects, the lymphocyte distribution values were significantly higher than in unexposed subjects (*p* = 0.02). In male subjects exposed to formaldehyde, mean monocyte values were significantly higher than in unexposed subjects (6.5% vs. 3.1%; *p* = 0.000); In male subjects exposed to formaldehyde, the distribution of monocytes was significantly higher than in unexposed subjects (*p* = 0.000). A study of nurses in Taiwan showed that formaldehyde exposure was correlated with reduced white blood cell counts [41,42]. A recent study in China showed that formaldehyde was associated with lowered T cells in the blood of exposed workers [43]. Several studies in the Chinese literature have also reported that occupational exposure to formaldehyde was associated with a decrease in white blood cell counts and possibly other cell counts, such as platelets [44]. In our study, on the other hand, changes result for all CBC parameters except for lymphocytes, red blood cells, and HB, which instead remain unchanged. This leads to the observation that the results are not always unambiguous, because in some experiments a decrease in leukocytes was recorded after exposure to different irritants such as formaldehyde [45]. The explanation could stem from a phenomenon of hormesis, considered an adaptive function of the organism, whereby many biological systems exposed to a wide range of agents show opposite responses depending on the dose [46].

The strength of this study is the lack of studies in literature evaluating CBC parameters, whose hypotheses are consolidated by significant analysis. Our work is important to evaluate how acrylamide and formaldehyde caused similar changes in blood count values in workers and this conclusion is a novelty in the literature. Weaknesses are represented by the single-center data: the cohort of employees belongs only to the University of Rome La Sapienza. In addition, important data, such as time and years of exposure to chemical risk of the employees involved, are missing. However, there are several articles in which the authors used a dichotomous variable for assessing the exposure (exposed vs. not exposed) using a qualitative approach [47,48,49,50].

## 5. Conclusions

The present study is a cross-sectional study that evaluates the hemochromocytometric parameters in subjects exposed to chemical risks, in comparison with those observed in a population of non-exposed subjects, such as periodic blood donors, who can be considered as a healthy population, because they are periodically monitored by the Transfusion Centre Service to which the donors belong.

If the data of the present study are confirmed by prospective studies, it will be necessary to investigate the pathophysiological mechanisms by which exposure to chemicals determines the variations in hemochromocytometric parameters found in exposed versus unexposed individuals. The present study revealed a significant change in blood count values in subjects exposed to acrylamide and formaldehyde. The results also showed an increase in the average monocyte value in employees exposed to carcinogenic risk (except for Trypan Blue), which could suggest the hypothesis of a mechanism linked to a chronic inflammatory stimulus, supported by specific cytokines (IL-6, TNF, IL-10). The increase in efficient health surveillance, therefore, continues to be of fundamental importance in all work activities where there is a specific risk to employees.

## Figures and Tables

**Table 1 jcm-13-06317-t001:** Characteristics of the participants.

Parameter	Exposed to Carcinogens	Exposed to Chemicals	Non Exposed (Blood Donors)	*p*-Value
*Age* Mean +-SD	41.67 ± 14.59	39.48 ± 14.11	40.68 ± 13.87	0.206 * 0.017 ^
*Gender* Female Male	219 (61.5%) 137 (38.5%)	646 (61.8%) 400 (38.2%)	1890 (60.6%) 1231 (39.4%)	0.726 * 0.404 ^

* Comparison between exposed to carcinogens vs. not exposed. ^ Comparison between exposed to chemicals vs. not exposed.

**Table 2 jcm-13-06317-t002:** Comparison of average blood count parameters between chemically exposed and unexposed subjects.

Parameter	Exposed to Chemicals	Not Exposed (Blood Donors)	Significance
	Mean ± SD	Mean ± SD	*p*-Value *
WBC × 10^3^/µL	6.8 ± 1.8	6.4 ± 1.5	<0.001
Lymphocyte %	33.8 ± 7.5	36.8 ± 7.5	<0.001
Monocyte %	7.9 ± 1.9	6.7 ± 1.9	<0.001
PMN%	54.7 ± 8.7	56.4 ± 8.1	<0.001
Lymphocyte × 10^3^/µL	2.2 ± 0.6	2.2 ± 0.6	0.173
Monocyte × 10^3^/µL	0.5 ± 0.2	0.4 ± 0.1	<0.001
PMN × 10^3^/µL	3.8 ± 1.4	3.8 ± 1.2	0.995
Red blood cells × 10^6^/µL	4.9 ± 3.2	4.8 ± 0.4	0.27
Hb (g/dL)	13.9 ± 2.4	14.2 ± 1.2	<0.001
Ht (%)	41.7 ± 5.8	43.0 ± 3.8	<0.001
MCV (fL)	87.8 ± 8.1	90.3 ± 4.2	<0.001
MCH (pg)	29.4 ± 2.2	29.8 ± 1.6	<0.001
MCHC (g/dL)	33.1 ± 2.1	32.9 ± 1.0	0.002
Platelets × 10^3^/µL	238.5 ± 64.4	229.5 ± 51.7	<0.001
MPV (fL)	9.8 ± 30.7	8.2 ± 8.6	0.008

SD: standard deviation; * ANOVA test, statistical significance level <0.05.

**Table 3 jcm-13-06317-t003:** Comparison of average hemochromocytometric parameters between carcinogen-exposed and non-exposed subjects.

Parameter	Exposed to Carcinogens	Not Exposed (Blood Donors)	Significance
	Mean ± SD	Mean ± SD	*p*-Value *
PMN %	54.1 ± 8.6	56.7 ± 12.2	<0.001
Lymphocyte %	34.1 ± 7.4	37.0 ± 10.7	<0.001
Monocyte %	8 ± 1.7	6.7 ± 2.6	<0.001
WBC × 10^3^/µL	6.8 ± 1.7	6.4 ± 1.9	<0.001
PMN × 10^3^/µL	3.7 ± 1.4	3.8 ± 1.2	0.561
Lymphocyte × 10^3^/µL	2.3 ± 0.6	2.3 ± 0.6	0.945
Monocyte × 10^3^/µL	0.5 ± 0.2	0.4 ± 0.2	<0.001
Red blood cells × 10^6^/µL	56.2 ± 480.1	5.0 ± 10.5	<0.001
Hb (g/dL)	14.4 ± 7.0	14.2 ± 1.2	0.051
Ht (%)	43.3 ± 19.0	43.0 ± 3.8	0.551
MCV (fL)	86.6 ± 10.9	90.3 ± 4.2	<0.001
MCH (pg)	30.9 ± 20.1	29.8 ± 1.6	0.003
MCHC (g/dL)	33.6 ± 16.9	32.9 ± 1.0	0.024
Platelets × 10^3^/µL	234.6 ± 74.1	229.7 ± 51.8	0.112
MPV (fL)	10.4 ± 37.3	8.2 ± 8.6	0.005

SD: standard deviation; * ANOVA test, statistical significance level <0.05.

**Table 4 jcm-13-06317-t004:** Comparison between exposure to each individual carcinogen and non-exposure.

	Not Exposed	Acrylamide	Benzene	Formaldehyde	K Dichromate	Ligroin	Trypan Blue	Trichloroethylene	Other Carcinogens
	Mean ± DS	Mean ± DS	Mean ± DS	Mean ± DS	Mean ± DS	Mean ± DS	Mean ± DS	Mean ± DS	Mean ± DS
	*p*-Value *	*p*-Value *	*p*-Value *	*p*-Value *	*p*-Value *	*p*-Value *	*p*-Value *	*p*-Value *	*p*-Value *
PMN %	56.52 ± 8.09	54.73 ± 8.23	54.2 ± 7.94	55.02 ± 10.2	50.45 ± 9.39	56.37 ± 9.84	55.92 ± 8.13	45.9 ± 8.8	53.77 ± 8.03
		*p* = 0.57	*p* = 1.000	*p* = 1.000	*p* = 0.107	*p* = 1.000	*p* = 1.000	*p* = 0.008	*p* = 0.028
Lymphocyte %	36.81 ± 7.5	33.75 ± 7.34	34.49 ± 7.62	33.38 ± 8.37	36.08 ± 7.06	31.57 ± 8.53	34 ± 6.8	39.59 ± 7.17	34.51 ± 6.9
		*p* < 0.001	*p* = 1.000	*p* = 0.066	*p* = 1.000	*p* = 0.325	*p* = 1.000	*p* = 1.000	*p* = 0.079
Monocyte %	6.66 ± 1.88	7.97 ± 1.65	7.47 ± 1.45	8.03 ± 1.69	8.2 ± 1.48	7.86 ± 2.09	6.9 ± 1.59	9.13 ± 3.36	8.1 ± 1.75
		*p* < 0.001	*p* = 0.451	*p* < 0.001	*p* = 0.037	*p* = 0.588	*p* = 1.000	*p* = 0.007	*p* < 0.001
WBC × 10^3^/µL	6.4 ± 1.53	6.77 ± 1.68	7.37 ± 2.13	6.58 ± 1.51	7.18 ± 1.43	6.95 ± 2.43	6.48 ± 1.62	6.53 ± 0.96	6.68 ± 1.71
		*p* = 0.329	*p* = 0.011	*p* = 1.000	*p* = 1.000	*p* = 1.000	*p* = 1.000	*p* = 1.000	*p* = 1.000
PMN × 10^3^/µL	3.76 ± 1.22	3.74 ± 1.29	4.07 ± 1.65	3.66 ± 1.35	3.63 ± 1.04	4.05 ± 1.92	3.71 ± 1.39	2.99 ± 0.77	3.65 ± 1.42
		*p* = 1.000	*p* = 1.000	*p* = 1.000	*p* = 1.000	*p* = 1.000	*p* = 1.000	*p* = 1.000	*p* = 1.000
Lymphocyte × 10^3^/µL	2.26 ± 0.59	2.24 ± 0.69	2.45 ± 0.61	2.12 ± 0.65	2.57 ± 0.65	2.11 ± 0.63	2.14 ± 0.45	2.58 ± 0.62	2.26 ± 0.51
		*p* = 1.000	*p* = 1.000	*p* = 1.000	*p* = 1.000	*p* = 1.000	*p* = 1.000	*p* = 1.000	*p* = 1.000
Monocyte × 10^3^/µL	0.37 ± 0.14	0.53 ± 0.16	0.55 ± 0.21	0.51 ± 0.15	0.58 ± 0.11	0.53 ± 0.17	0.43 ± 0.1	0.58 ± 0.17	0.54 ± 0.18
		*p* < 0.001	*p* < 0.001	*p* < 0.001	*p* < 0.001	*p* = 0.001	*p* = 1.000	*p* = 0.002	*p* < 0.001
Red blood cells × 10^6^/µL	4.77 ± 0.43	4.61 ± 0.4	4.79 ± 0.41	4.65 ± 0.44	4.86 ± 0.48	4.92 ± 0.57	4.56 ± 0.36	4.97 ± 0.62	4.76 ± 0.47
		*p* = 0.001	*p* = 1.000	*p* = 1.000	*p* = 1.000	*p* = 1.000	*p* = 1.000	*p* = 1.000	*p* = 1.000
Hb (g/dL)	14.18 ± 1.17	13.42 ± 1.14	13.83 ± 1.15	13.65 ± 1.33	14.19 ± 1.43	14.12 ± 0.84	12.82 ± 1.1	14.75 ± 1.64	13.84 ± 1.42
		*p* < 0.001	*p* = 1.000	*p* = 0.087	*p* = 1.000	*p* = 1.000	*p* = 0.021	*p* = 1.000	*p* = 0.15
Ht (%)	43.05 ± 3.77	40.35 ± 3.33	42.27 ± 3.46	40.82 ± 3.8	42.82 ± 3.94	43.15 ± 2.93	38.96 ± 3.26	44.2 ± 4.37	41.9 ± 4.1
		*p* < 0.001	*p* = 1.000	*p* = 0.002	*p* = 1.000	*p* = 1.000	*p* = 0.04	*p* = 1.000	*p* = 0.085
MCV (fL)	90.36 ± 4.25	87.76 ± 6.22	88.4 ± 4.65	88.02 ± 5.97	88.05 ± 2.62	88.24 ± 5.53	85.84 ± 9.43	89.21 ± 3.28	88.32 ± 6.39
		*p* < 0.001	*p* = 0.39	*p* = 0.015	*p* = 1.000	*p* = 1.000	*p* = 0.088	*p* = 1.000	*p* < 0.001
MCH (pg)	29.77 ± 1.6	29.32 ± 2.46	29.12 ± 1.69	29.47 ± 2.26	29.17 ± 0.92	28.93 ± 2.14	28.3 ± 3.46	29.7 ± 1.24	29.2 ± 2.24
		*p* = 0.115	*p* = 0.852	*p* = 1.000	*p* = 1.000	*p* = 1.000	*p* = 0.311	*p* = 1.000	*p* = 0.023
MCHC (g/dL)	32.95 ± 1.01	33.29 ± 1.21	32.94 ± 1.23	33.49 ± 1.18	33.11 ± 1.36	32.78 ± 1.08	32.98 ± 1.75	33.3 ± 0.91	33.07 ± 1.01
		*p* = 0.013	*p* = 1.000	*p* = 0.015	*p* = 1.000	*p* = 1.000	*p* = 1.000	*p* = 1.000	*p* = 1.000
Platelets × 10^3^/µL	230.79 ± 53.3	248.25 ± 64.85	244.59 ± 104.31	235 ± 82.49	226.19 ± 52.84	284.79 ± 154.53	223.56 ± 44.85	206.13 ± 30.88	227.53 ± 49.75
		*p* = 0.021	*p* = 1.000	*p* = 1.000	*p* = 1.000	*p* = 0.01	*p* = 1.000	*p* = 1.000	*p* = 1.000
MPV (fL)	7.85 ± 0.77	8.47 ± 1.1	8.18 ± 0.81	8.72 ± 1.01	8.31 ± 0.83	8.21 ± 1.57	8.73 ± 1.3	8.16 ± 0.64	8.44 ± 1.04
		*p* < 0.001	*p* = 0.605	*p* < 0.001	*p* = 0.728	*p* = 1.000	*p* = 0.033	*p* = 1.000	*p* < 0.001

SD: standard deviation; * ANOVA test, statistical significance level <0.05.

## Data Availability

Data supporting reported results can be requested to the authors.

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
