# Peer review of "Does Occupational Exposure to Chemicals/Carcinogens Affect the Hematological Parameters of Workers?"

_jcm, 2024, doi:10.3390/jcm13216317_

Round 1

Reviewer 1 Report

Comments and Suggestions for Authors

In this manuscript "Does occupational exposure to chemicals/ carcinogens. affect the hematological parameters of workers?", a team led by Torre and Sernia, describe an asscoiation between exposure to chemical risk and changes in haematochemical parameters of workers exposed. The author enrolled 4523 employees vs a cohort of 3121 blood donors.

The author made several interesting points here, and present in well structured form. There were several weaknesses that need to addressed.

1. Some sentences from line 22 to line 27 might need to be rewritten to read more smoothly.

2. References are needed in Line 36-line 40, line 61-line 67, and line 68-line 72.

3. Are all the donors in the control group healthy? Was there no exposure in the control group?

4. Many similar papers have been published. The author needs to clarify the novelty of their work.

5. What about other parameters, such as immunological parameters?

Reviewer 2 Report

Comments and Suggestions for Authors

jcm-3191954

Giuseppe La Torre et al. report an association between an exposure to chemicals and changes in hematological parameters of 1402 workers exposed in the University during 16 months. They concluded acrylamide and formaldehyde caused similar changes in blood count values in workers. I think they reported interesting findings, but several concerns can be listed.

1) There are too many data on hematology alterations in workers who exposed chemicals. They are complicated and it is hard to follow. Please specify chemicals and/or carcinogens, except for acrylamide and formaldehyde.

2) In addition, exposure levels of each chemical can be described.

3) Please describe what kind of exposure, airborne or contact. Please identify departments of workers belonging to (pathology, anatomy labs or operating rooms).

4) If possible, please use figures or illustrations made from the results obtained for us to easy understand.

5) Tables: “Lynfocyte” must be “Lymphocyte”.

Round 2

Reviewer 1 Report

Comments and Suggestions for Authors

The authors’ responses in the revised manuscript mostly addressed those critiques adequately. The resulting manuscript is much improved.

Author Response

Many thanks for your comments and suggestions

Reviewer 2 Report

Comments and Suggestions for Authors

The revision is not enough for publication. As the authors stated in the Discussion section, important data, such as the duration of chemical exposure is lacking. This is a fatal flow in this type of study. In addition, the title can be modified: “carcinogen” should be deleted. All chemicals exposed to workers should be described in the text.
